# Debiasing Algorithm through Model Adaptation

**Tomasz Limisiewicz**      **David Mareček**      **Tomáš Musil**
Faculty of Mathematics and Physics, Charles University
{limisiewicz,marecek,musil}@ufal.mff.cuni.cz

## Abstract

Large language models are becoming the go-to solution for the ever-growing number of tasks. However, with growing capacity, models are prone to rely on spurious correlations stemming from biases and stereotypes present in the training data. This work proposes a novel method for detecting and mitigating gender bias in language models. We perform causal analysis to identify problematic model components and discover that mid-upper feed-forward layers are most prone to convey bias. Based on the analysis results, we intervene in the model by applying a linear projection to the weight matrices of these layers. Our titular method *DAMA* significantly decreases bias as measured by diverse metrics while maintaining the model's performance on downstream tasks. We release code for our method and models, which retrain *LLaMA*'s state-of-the-art performance while being significantly less biased.[1]

## 1 Introduction

Large language models have a large capacity for learning linguistic and factual information from training data, but they are prone to capture unwanted biases. It has been shown that LLMs are gender biased (Stanczak & Augenstein, 2021; Blodgett et al., 2020; van der Wal et al., 2023; Nadeem et al., 2021; Nangia et al., 2020; Limisiewicz & Mareček, 2022). This bias is manifested by relying on a spurious correlation between seemingly gender-neutral expressions and specific gender. For instance, language models tend to ascribe stereotypical gender to certain practitioners, e.g. by outputting high probabilities for phrases such as "male mechanics" or "female cleaners" (Lu et al., 2020b). In many tasks, the models also show uneven performance for the test examples involving different gender contexts.

This work analyzes the *LLaMA* family of models (Touvron et al., 2023). These openly available models obtain state-of-the-art performance on a variety of downstream tasks. We focus specifically on the gender bias present in these models, but our method is applicable to other types of bias as well. We specifically ask: 1) Can we identify evidence of gender bias in *LLaMA*? Specifically, do they associate professional names with the stereotypical gender? 2) Can we identify which components of the model store the gender-biased representation? 3) Can we edit the model's weights to decrease the bias while preserving its performance on end-tasks?

To answer the first question, we check the *LLaMA* performance on popular tests for gender bias: WinoBias (Zhao et al., 2018) and StereoSet (Nadeem et al., 2021). We introduce an interpretable metric that evaluates bias on the language generation task. To answer the second question, we perform causal tracing (Vig et al., 2020; Meng et al., 2022a). We monitor changes in the distribution of predictions when the stereotypical representation is revealed only in one of the components, such as MLP (multilayer perceptron) or attention layer. Following the terminology of Pearl (2001), we call such component *gender bias mediator*. To tackle the last question, we introduce "***D**ebiasing **A**lgorithm through **M**odel **A**daptation*". In *DAMA*, we edit bias-vulnerable feed-forward layers by multiplying linear transformation weights by the orthogonal projection matrix similar to Ravfogel et al. (2022). Our results show that with directed changes in model weights, we can reduce gender bias substantially while having only a minimal impact on the model's performance. Specifically,

---

[1]The code available at: github.com/tomlimi/DAMA

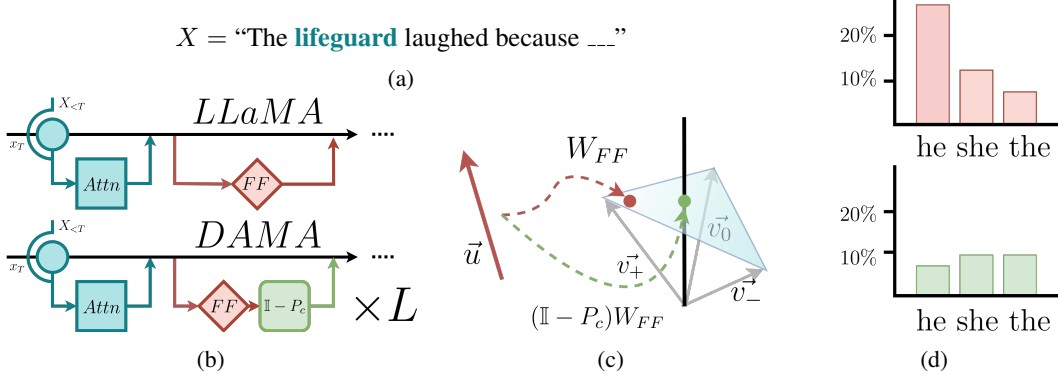

Figure 1: Schema (b) shows *DAMA* intervention in a LLaMA layer. Even though $\mathbb{I} - P_c$ is depicted as a separate module, in practice, it is multiplied with the output matrix of a feed-forward layer ($W_{FF}$). Therefore, *DAMA* is neutral to the model's parameter count and architecture. (a) We show the behavior of the model when presented with a stereotypical prompt. Specifically, (c) shows the projections of the feed-forward latent vector ($\vec{u}$) onto the output space. With *DAMA* (lower arrow), we nullify the gender component of the representation. It results in balanced probabilities of gendered tokens in the model's output, as shown in (d).

we monitor performance changes in language modeling (measured by perplexity) and in four downstream tasks.

To list our contributions: We evaluate gender bias in *LLaMA* models and introduce a novel, transparent metric for quantifying bias directly in language generation. Most importantly, we propose *DAMA*, a method for editing weights of the bias mediator to significantly reduce gender bias in three different tasks without sacrificing performance across unrelated tasks. This is an improvement over prior methods that were focused on one type of bias manifestation (Ranaldi et al., 2023) or were not tested for preserving language understanding capabilities of the model (Lauscher et al., 2021; Gira et al., 2022).

## 2 METHODOLOGY AND EXPERIMENTAL SETUP

### 2.1 LLAMA MODELS

*LLaMA* models are causal language models following Transformer decoder architecture (Vaswani et al., 2017). *LLaMA* family contains models with 7B, 13B, 30B, and 65B parameters. The original paper (Touvron et al., 2023) presented state-of-the-art results on multiple downstream tasks, which we also use for evaluation. In our implementation, we used the model checkpoint accessible through the Huggingface library `huggingface.co`. Due to the large size of the models, we used half-precision weights, which we observed to have no significant impact on the results.

### 2.2 GENDER BIAS EVALUATION IN LANGUAGE GENERATION

To better understand gender bias in language generation, we construct our dataset of prompts and an interpretable diagnostic measure.

We use the set of professions chosen and annotated by Bolukbasi et al. (2016).[2] Each profession was assigned two scores: *factual* score $x_f$ (originally called *definitionality*) and *stereotypical* score $x_s$. They define how strongly a word is connected with the male or female gender respectively through semantically or through stereotypical cues. By convention, scores range from $-1$ for

---

[2]The data is available at: `https://github.com/tolga-b/debiaswe/blob/master/data/professions.json`

female-associated words to 1 for male ones.[3] We fill the proposed profession words in the prompts of the structure presented in Figure 1a. The **lifeguard** is, by definition, a gender-neutral word ($x_f = 0$) and associated with the male gender by a stereotypical cue ($x_s = 0.6$). We measure the probabilities for gendered prediction for a given prompt $P_M(o|X)$. For that purpose, we use pronouns $o_+ = $ "he" and $o_- = $ "she", as they are probable continuations for given prompts.

Subsequently for each prompt, we compute *empirical* score $y = P_M(o_+|X) - P_M(o_-|X)$. To estimate the relationship between the observed score and annotated ones $x_s$ and $x_f$, we construct a linear model:

$$y = a_s \cdot x_s + a_f \cdot x_f + b_0 \tag{1}$$

The linear fit coefficients have the following interpretations: $a_s$ is an impact of stereotypical signal on the model's predictions; $a_f$ is an impact of the factual (semantic) gender of the word. Noticeably, $y$, $x_s$, and $x_f$ take the values in the same range. The slope coefficient tells how shifts in annotated scores across professions impact the difference in prediction probabilities of male and female pronouns. The intercept $b_0$ measures how much more probable the male than the female pronouns are when we marginalize the subject. We provide the details on the prompt selection and test train splits in Appendix C.

## 2.3 OTHER GENDER BIAS INDICATORS

We also evaluate other well-established methods of evaluating gender bias manifestations in two downstream tasks:

**WinoBias**  Zhao et al. (2018) present the dataset containing a WinoGrad scheme (Levesque et al., 2011) examples. Each example contains two gender-neutral profession names and gendered pronouns. The task is to identify the coreference link between the pronouns and the correct professional. The dataset consists of two parts: pro-stereotypical, where coreference links to a profession name with stereotypical gender matching the gender of the pronoun; in anti-stereotypical examples, the profession's stereotypically assumed gender is different from the gender of the pronouns. An example from the anti-stereotypical subset: *"The nurse notified the patient that **his** shift would be ending in an hour."*[4]

We evaluate the general accuracy of coreference resolution *Acc*. Inspired by the analysis of Stanovsky et al. (2019), we introduce two measures of bias: $\Delta S$: the difference between the accuracy of pro and anti-stereotypical examples, and $\Delta G$: the difference between the accuracy between examples containing male and female pronouns.

**StereoSet**  Nadeem et al. (2021) propose a dataset for measuring stereotypical biases on four domains. We consider only *gender* domain and *intrasentence* task. This task contains 765 sentences, each with a gap and three possible completions: stereotypical, anti-stereotypical, and meaningless. E.g., a sentence *Girls tend to be more _____ than boys.* and three completions *soft*, *determined*, and *fish*. The authors propose three evaluation measures: 1) *lms* – the percentage of sentences where the model prefers the meaningful over the meaningless completion; 2) *ss* – the percentage of sentences where the model prefers the stereotypical over the anti-stereotypical completion; and 3) *icat* score that combines the previous two: $icat = lms \cdot \min(ss, 100 - ss)/50$. Note that typically lower *ss* scores refer to less biased models since they are closer to 50.

## 2.4 LANGUAGE MODELING

To evaluate the performance of the model's pre-training task, we measure perplexity on the Wikipedia 103 corpus (Merity et al., 2016) available through HuggingFace.

---

[3]We use positive values for male gender following the original paper. This is only an arbitrary choice, and switching polarities wouldn't affect this analysis. Importantly, we do not intend to ascribe negative valuations to any of the genders.

[4]In this example, the coreferential link relies on semantics, while in other instances, coreference can be resolved solely through syntax.

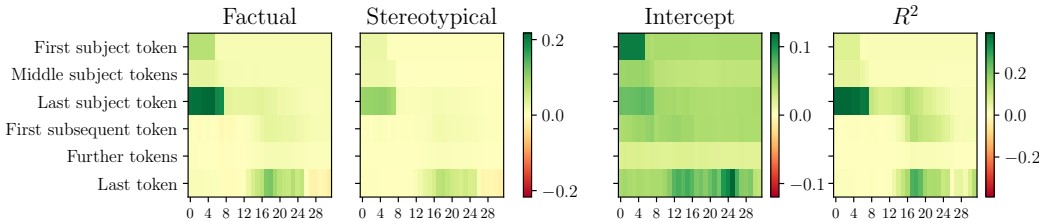

Figure 2: Causal tracing of *factual* $a_f$, *stereotypical* $a_s$ coefficients and *intercept* $b$ in regression to indirect effects of the model $y_{IE}$. The linear models are independently fitted for restored **MLP** *clean* representation at each layer and token position.

## 2.5 Downstream Tasks

We have selected three datasets that measure common sense reasoning and language understanding to evaluate the possible performance loss after altering the model: **OpenBookQA (OBQA)** (Mihaylov et al., 2018) contains 500 multiple-choice questions aimed at combining science facts with common knowledge. **AI2 Reasoning Challenge (ARC)** (Clark et al., 2018) contains natural science questions authored for use on standardized tests. It is partitioned into a Challenge Set (1172 test questions) and an Easy Set (2376 test questions). **Massive Multitask Language Understanding (MMLU)** (Hendrycks et al., 2021) contains 14 042 questions on 57 topics, including math, law, or social sciences. The former two tasks are evaluated in a zero-shot regime. In the MMLU, we provide five in-context examples. In all the evaluations, we followed closely the original setting of Touvron et al. (2023).

## 3 Bias Evaluation and Causal Tracing

### 3.1 Experiments

**Bias Evaluation**   We assess gender bias in *LLaMA* by employing the linear model outlined in Section 2.2. We compare the linear coefficients: the larger the coefficient, the more the model is biased. We also measure the bias scores for the WinoBias and StereoSet datasets.

**Causal Tracing**   To identify the components storing gendered associations, we perform causal tracing for gender bias in text generation. We use a similar methodology as Meng et al. (2022a). For each test prompt, (1) we perform a *clean run* and collect all the activations at all layers and tokens; (2) we perform a *corrupted run* by adding noise to the tokens of the profession (details in Appendix C ); (3) we perform *corrupted runs* with restoration: at each step, we restore the activations from the *clean run* of each output of MLP at one particular layer and token. For each layer $l$, token position $i$, and a prompt $X$ we compute the score $y_{l,i}(X) = P_{l,i}(o_+|X) - P_{l,i}(o_-|X)$. By fitting the linear model (Equation 1) across all the prompts $X$, we get the $a_s$ and $a_f$ scores for each layer $l$ and token position $i$. Following Meng et al. (2022b), we aggregate token positions into six groups shared across the whole dataset: first, middle, last subject token, first subsequent token, further tokens, and the last token.

### 3.2 Results

**Bias Evaluation**   We show the coefficient of the linear model in Table 1. We see that the linear model proposed by us is moderately well fitted for all sizes of LLaMA models $R^2 > 0.35$. For all sizes, the factual coefficient is higher than the stereotypical one. The models are more influenced by semantic than stereotypical cues ($a_f > a_s$). Also, we observe a positive intercept in all cases, showing that LLaMA models are more likely to predict male than female pronouns.

Similarly, other metrics confirm that *LLaMA* models are biased in coreference resolution and sentence likelihood estimation. In WinoBias, we observe that the bias stemming from stereotypes $\Delta S$ is more prominent than the accuracy difference between examples with male and female pronouns $\Delta G$.

| | Bias in LM | | | | WinoBias | | | StereoSet gender | | |
|---|---|---|---|---|---|---|---|---|---|---|
| | $\downarrow a_s$ | $\uparrow a_f$ | $\downarrow b$ | $\downarrow R^2$ | $\uparrow$ Acc | $\downarrow \Delta S$ | $\downarrow \Delta G$ | $\uparrow$ lms | $\downarrow$ ss | $\uparrow$ ICAT |
| MEMIT | 0.209 | 0.282 | 0.071 | 0.497 | 59.3% | 40.5% | 3.3% | 95.6 | 72.0 | 53.6 |
| LoRA FT | 0.144 | 0.261 | -0.040 | 0.413 | 58.8% | 34.4% | 5.6% | 89.0 | 62.9 | 66.0 |
| LLaMA 7B | 0.235 | **0.320** | 0.072 | 0.494 | **59.1%** | 40.3% | 3.0% | 95.5 | 71.9 | 53.7 |
| DAMA | **-0.005** | 0.038 | **-0.006** | **0.208** | 57.3% | **31.5%** | 2.3% | 95.5 | 69.3 | **58.5** |
| ± (std) | 0.004 | 0.004 | 0.004 | 0.026 | 0.5% | 0.9% | 0.7% | 0.3 | 0.8 | 1.5 |
| LLaMA 13B | 0.270 | 0.351 | 0.070 | 0.541 | 70.5% | 35.7% | -1.5% | 95.2 | 71.4 | 54.4 |
| DAMA | 0.148 | 0.222 | 0.059 | 0.472 | 66.4% | 31.1% | -1.1% | 94.4 | 68.6 | 59.4 |
| LLaMA 30B | 0.265 | 0.343 | 0.092 | 0.499 | 71.0% | 36.0% | -4.0% | 94.7 | 68.4 | 59.9 |
| DAMA | 0.105 | 0.172 | 0.059 | 0.471 | 63.7% | 26.7% | -3.7% | 94.8 | 65.7 | 65.0 |
| LLaMA 65B | 0.249 | 0.316 | 0.095 | 0.490 | 73.3% | 35.7% | 1.4% | 94.9 | 69.5 | 57.9 |
| DAMA | 0.185 | 0.251 | 0.100 | 0.414 | 71.1% | 27.2% | 0.8% | 92.8 | 67.1 | 61.1 |

Table 1: Bias evaluation for the *LLaMA* models and their debiased instances Significance analysis for the 7B model was performed by running *DAMA* with five random seeds. We bold the score for the original model or *DAMA*, whichever is better if there are more than two standard deviations apart. We underline the best value in each column.

| | LM | Downstream | | | |
|---|---|---|---|---|---|
| | $\downarrow$ PPL | $\uparrow$ ARC-C | $\uparrow$ ARC-E | $\uparrow$ OBQA | $\uparrow$ MMLU |
| MEMIT | 26.1 | 42.7 | 68.9 | 57.0 | 30.2 |
| LoRA FT | 51.1 | 37.7 | 66.5 | 45.6 | 26.6 |
| LLaMA 7B | **26.1** | 42.2 | **69.1** | 57.2 | 30.3 |
| DAMA | 28.9 | 41.8 | 68.3 | 56.2 | 30.8 |
| ± (std) | 0.2 | 0.4 | 0.2 | 0.5 | 0.5 |
| LLaMA 13B | 19.8 | 44.9 | 70.6 | 55.4 | 43.3 |
| DAMA | 21.0 | 44.7 | 70.3 | 56.2 | 43.5 |
| LLaMA 30B | 20.5 | 47.4 | 72.9 | 59.2 | 55.7* |
| DAMA | 19.6 | 45.2 | 71.6 | 58.2 | 56.1* |
| LLaMA 65B | 19.5 | 44.5 | 73.9 | 59.6 | —* |
| DAMA | 20.1 | 40.5 | 67.7 | 57.2 | —* |

Table 2: Performance evaluation for the *LLaMA* models and their debiased instances. The significance analysis was performed the same as in Table 1. (*) Due to hardware limitations, we could not run MMLU inference for 65B models. In the evaluation of 30B model, we excluded 4% longest prompts.

**Causal Tracing**  In Figure 2, we observe the indirect effect of MLPs in each layer and token position of the 7B model. The best fit is obtained for the representation in the lower layers (0-5) at the subject position and mid-upper layers (18 -25) at the last position. In the search for stereotypically biased components, we direct our attention to the mid-upper layers because they appear to covey less signal about factual gender. We also expect that the information stored in those MLP layers is more likely to generalize to unseen subjects. Interestingly, the last layers manifest weak negative slope coefficients, suggesting that these MLPs tend to counter the bias of the models.

In Figure 4 (in Appendix B), we show the results of casual tracing for attention and the whole layer. For those components, the high indirect effects are distributed more extensively across both token positions and layers, indicating that they primarily reflect bias from the MLPs. For larger models, we observe analogous patterns shifted according to the total layer count.

## 4 DEBIASING ALGORITHM THROUGH MODEL ADAPTATION

We introduce the algorithm that decreases bias in language models by directly editing the model weights. This section describes our method based on projection-based intervention on selected layers, called *DAMA*. Further, we provide theoretical and empirical backing for the method's effectiveness.

### 4.1 Obtaining Stereotype Keys and Gendered Values

Following the convention from Geva et al. (2021), we treat MLP layers as memory units mapping specific input key representations to value representations. Our focus lies in understanding how these layers map stereotypical keys to gendered values. As our choice of keys, we take prompts introduced in Section 2.2, which carry stereotypical signal. The values are the output vectors corresponding to one of the personal pronouns (male, female, or neutral).

To compute the stereotypical key at $l$th layer, we feed the stereotypical prompt $X$ up to $l$ layer's feed-forward MLP ($FF_l$) to obtain its vector representation. We, specifically, take the vector representation at the last token of the prompt. We denote stereotypical keys as $u \in \mathbb{R}^{d_{FF}}$ following the convention from Figure 1c.

To compute the value representation corresponding to a specific gender, we employ the next-token prediction task based on the stereotypical prompt $X$. As possible next token, we consider one of the pronouns indicating gender ($O_+ = $ "$he''$" for male, $O_- = $ "$she''$" for female, and $O_0 = $ "$they''$" for neutral). We use the regular cross-entropy loss and optimize the output of the $l$th layer's feed-forward denoted $\mathcal{V}$:

$$v_o = \underset{z \in \mathbb{R}^{d_M}}{\arg\min} \left[ -\log P_{M[\mathcal{V}=z]}(o|X) + \lambda_1 D_{KL}[P_{M[\mathcal{V}=z]}(o'|X')||P_M(o'|X')] + \lambda_2||z||^2 \right] \quad (2)$$

The second part of the loss is added to preserve the model's LM capabilities for predicting the next token ($o'$) given general (not-biased) prompts ($X'$). The last summand is $L2$ regularization. We use gradient descent with 20 iterations to obtain a value vector for each of the pronouns $v_o \in \mathbb{R}^{d_M}$.

### 4.2 Obtaining Projection on Stereotype Subspace with PLS

To identify the stereotype subspace, we concatenate value vectors for each pronoun (male, neutral, and female) across all prompts to obtain gendered value matrices $V_+$, $V_0$, and $V_-$. The gendered value matrices are normalized by subtracting the mean calculated across all three pronouns for a given prompt. Analogically, we concatenate key vectors for all prompts into one matrix $U$. Then, we multiply it by the feed-forward's output matrix denoted $W_{FF,out,l}$:

$$W_{FF,out,l} \cdot U \to \hat{U} \quad (3)$$

We concatenate $V_+$, $V_0$, and $V_-$ together and concatenate $\hat{U}$ three times. We use the Partial Least Squares algorithm to identify the linear mapping $B_1$ maximizing correlation between stereotypical keys $[\hat{U}, \hat{U}, \hat{U}]$ and gendered values $[V_+, V_0, V_-]$:

$$[V_+, V_0, V_-] \approx_{\text{PLS}} B_1 \cdot [\hat{U}, \hat{U}, \hat{U}] + B_0 \quad (4)$$

By definition of PLS, $B_1$ identifies the stereotypical directions most correlated with gendered values.[5] Therefore, we compute the matrix projecting representation on subspace orthogonal to the one spanned by $d_c$ first columns of $B_1$ to nullify the stereotypical signal. For brevity, we denote the trimmed matrix as $B_1^{d_c} = B_1[:, :d_c]$. The projection is given by the equation:

$$P = \mathbb{I} - P_c = \mathbb{I} - B_1^{d_c}(B_1^{d_c T}B_1^{d_c})^{-1}B_1^{d_c T} \quad (5)$$

Finally, we perform the model editing by multiplying $l$th MLP feed-forward matrix $W_{FF,out,l}$ by the projection matrix $P$, see Figure 1c. Our algorithm *DAMA* is based on iterative computation and applying projections to feed-forwards of multiple subsequent MLP layers. It changes neither the model's architecture nor parameter sizes, as the result of matrix multiplication is of the same dimensionality as the original feed-forward matrix.

### 4.3 Theoretical Perspective

In this section, we show theoretical guarantees that multiplying linear feed-forward matrix $W_{FF,out,l}$ by projection matrix $P$ will be the optimal mapping between keys ($U$) and values ($V$), fulfilling that $W_{FF,out,l} \cdot U$ is orthogonal to the guarded bias subspace $\mathcal{C}$.

---

[5]Matrix $B_0$ can be used to normalize the value matrix. However, we have noticed that its loadings become nearly zero due to the earlier normalization of $[V_+, V_0, V_-]$.

**Theorem 1.** *Assume that we have a linear subspace $\mathcal{C} \subseteq \mathbb{R}^o$. Given a n-element key matrix $U \in \mathbb{R}^{i \times n}$ a value matrix $V \in \mathbb{R}^{o \times n}$, we search a mapping matrix $W \in \mathbb{R}^{o \times i}$ minimizing the least squares and satisfying $\forall_{i=1}^{n} W u_i \perp \mathcal{C}$. Specifically, we solve:*

$$\hat{W} = \arg\min_{W} ||WU - V||_F^2 \quad \text{such that} \quad \forall_{i=1}^{n} W u_i \perp \mathcal{C}$$

*This equation is solved by:*

$$\hat{W} = (\mathbb{I} - P_c)VU^T(UU^T)^{-1}$$

*Where $P_c$ is a projection matrix on a subspace $\mathcal{C}$.*

The proof of the theorem is in Appendix A. Noteworthy $VU^T(UU^T)^{-1}$ solves the regular mean square error problem of mapping prompt keys to values corresponding to the model's output. Due to gradient optimization in the model's pre-training, we can assume that in general case $W_{FF,out,l} = VU^T(UU^T)^{-1}$. Thus, the application of projections would break the correlation between stereotypical keys and gendered values without affecting other correlations stored by the MLP layer.

### 4.4 EMPIRICAL PERSPECTIVE

**Effectivness** We apply *DAMA* to MLPs in approximately one-third of the model's upper layers (in *LLaMA* 7B layers 21 - 29 out of 32 with projection dimensionality $d_c = 256$). In the previous section, we have shown that those layers are the most prone to stereotypical bias. We check the impact of *DAMA* on bias coefficients of linear model (see Section 2.2) and LM perplexity. Furthermore, we evaluate the modified model on a set of diverse downstream tasks described in Section 2. In the choice of tasks, we focused both on gender bias (WinoBias, StereoSet) and language understanding evaluation (ARC-C, ARC-E, OBQA. MMLU).

**Baselines** We compare the method with a similar model editing method **MEMIT** (Meng et al., 2023) and a parameter-efficient fine-tuning via **LoRA** (Hu et al., 2022). In both baselines, we optimize by the objective of predicting a randomly sampled pronoun when presented with a biased prompt.

**Choice of Layers and Dimensionality** We analyze how the results vary depending on the number of layers selected for debiasing Due to the iterative character of intervention, we always start editing at the fixed layer (22 in *LLaMA* 7B) and gradually add subsequent layers. Further, we check the effect of the number of projection dimensions ($d_c$) in the power sequence from 32 to 1024.

**Scaling** Lastly, we examine the algorithm's performance for larger scales of *LLaMA* model: 13B, 30B, and 65B.

### 4.5 RESULTS

**Effectivness** *DAMA* effectively decreases the gender bias of the model while preserving its performance on other tasks, as seen in Table 1. Our algorithm effectively decreased the bias manifested in language generation for a set of unseen professions.[6]

Morover, *DAMA* significantly mitigates bias in StereoSet and WinoBias. In the latter task, general accuracy decreases, presumably due to the weakening of the stereotypical cue contributing to correct predictions in numerous test examples.

Our observations confirm that MLP layers contain stereotypical correlations responsible for multiple manifestations of bias. Furthermore, we observe in Table 2 that the algorithm causes a slight deterioration in general language modeling measured by perplexity on Wikipedia texts. It has a minor reflection in performance for downstream tasks. The altered model achieves a slightly lower score, yet differences are statistically significant only for one task (ARC-E). Therefore, we can conclude that *DAMA* does not impact the model's ability in question-answering tasks.

---

[6]In Table 3, we also show examples of next token probabilities in the original and debiased model.

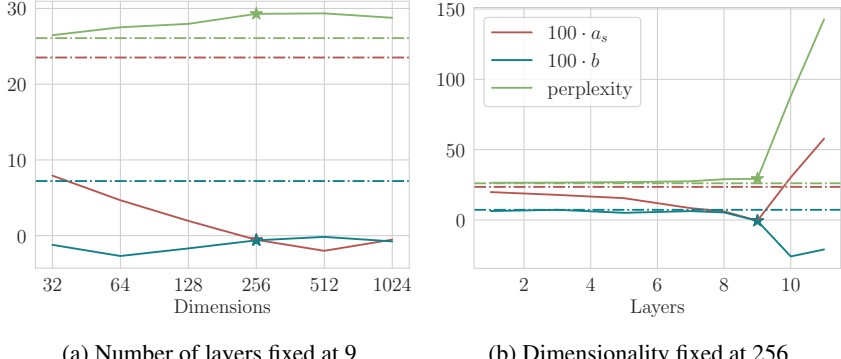

(a) Number of layers fixed at 9      (b) Dimensionality fixed at 256

Figure 3: The effect of applying *DAMA* to *LLaMA* 7B model on performance and bais in language modeling. We measured bias on gendered prompts (Section 2.2) by linear coefficients: $a_s$ and $b$ coefficient, the causal language modeling capabilities are measured by perplexity. Stars mark the performance of the model picked for further evaluation. The dashed line corresponds to the scores of the original *LLaMA* 7B model.

**Baselines** In contrast to *DAMA*, MEMIT has a minor effect on bias measures. We think it is because it is aimed to alter information specific to key-value pairs selected for intervention. Therefore, the intervention performed on the training set of professions does not generalize to unseen professions or other types s of gender bias. LoRA manifests stronger debiasing properties, coming close to the results of *DAMA* in multiple bias metrics, and performs better in StereoSet $ss$ and $ICAT$. Nevertheless, fine-tuning significantly deteriorates perplexity and the performance in language understanding tasks.

**Choice of Layers and Dimensionality** In Figure 3, we observe that the choice of the number of layers for debiasing and the dimensionality of projection affect both parameters. Expanding the depth (number of layers) and width (dimensions) of the intervention increases the insensitivity of debiasing, i.e., decreases $a_s$ and $b$ coefficients and negatively impacts perplexity. Interestingly, we observe a negative impact on both measured aspects when applying *DAMA* on the two last layers of the models. As noted in Section 3.1, the MLPs in those layers tend to counter bias in the original model.

**Scaling** We performed a coarse hyperparameter search for sensitive parameters of *DAMA*: number of layers and dimensionalities of the projections. Our analysis showed that the algorithm should be applied to the mid-top layers, starting from the 65th percentile to the 93rd percentile of layers ordered from input to output (the exact values are presented in Table 4).

We have achieved a notable reduction in bias scores for all models. Noticeably, although we do not observe the shared pattern for the bias metrics across different model sizes, the improvements brought by *DAMA* are consistent. Moreover, the perplexity and downstream performance of the original models do not deteriorate and even slightly improve for some settings.

## 5 DISCUSSION

Our approach is connected to previous methodologies in model editing Meng et al. (2022b) and bias mitigation (Ravfogel et al., 2022). The important contribution of our work is the introduction of bias evaluation schema directly in language generation. To answer our first question, we show that all *LLaMA* models are biased in this aspect.

Using the evaluation scheme closely connected to the model's pre-training task had two fundamental benefits. Firstly, it allowed us to perform a causal analysis of model components. The analysis allowed us to answer our second research question. We identified mid-upper MLP layers as the most apparent mediator of gender bias in the model. Secondly, we could perform debiasing adaptation directly on the model's weights without using a proxy task (Ravfogel et al., 2022) or fine-tuning on

limited data that often deteriorates the model's general performance (Gira et al., 2022). Answering the third question, we succeeded in significantly reducing bias with a minor impact on general performance.

The proposed algorithm generalizes the applicability of model-editing (Meng et al., 2022a;b; Mitchell et al., 2022; De Cao et al., 2021) to the case of modifying general dataset artifacts instead of the information specific to particular examples. Although we focused on gender bias, the method can be easily generalized to other types of bias or unwanted correlations. Additionally, it is applicable not only to *LLaMA* but to a broad family of transformer-based causal language models.

**Future Work**  We plan to improve the method of finding projection matrices, possibly using a convex search (Ravfogel et al., 2022) or analytically derived pseudo-projections (Belrose et al., 2023). We aim to investigate further the ranges of layers and dimensions that convey bias to apply *DAMA* on other model types effectively. Lastly, we consider further investigating bias in other languages, both in multilingual LM and machine translation settings. We are particularly interested in how our approach can be generalized for morphologically rich languages with more ubiquitous gender marking than English (Zmigrod et al., 2019).

## 6 RELATED WORK

**Measuring bias in language model**  Gender bias in language models has multiple manifestations quantified by various metrics, which often show low mutual correlation (Delobelle et al., 2022; van der Wal et al., 2023). One common approach to operationalize bias is to compare the probability assigned by a model to sentences conveying neutral and stereotypical information, e.g. SeteroSet (Nadeem et al., 2021), CrowS-Pairs (Nangia et al., 2020). Probability-based methods were criticized for being sensitive to the annotation choices (Blodgett et al., 2021) and are hard to apply to auto-regressive models such as *LLaMA*.

Another popular method to estimate gender bias is based on the coreference task, where personal pronouns should be assigned to the correct antecedent in Winograd scheme (Levesque et al., 2011), e.g. WinoBias (Zhao et al., 2018), Winogender (Rudinger et al., 2018). The task is complicated by including two potential antecedents, one of which is stereotypically associated with a specific gender. The analysis of such examples shows that models struggle with solving non-stereotypical links.

**Debiasing methods**  Similarly to the number of bias metrics, researchers proposed various debiasing methods (Stanczak & Augenstein, 2021; Savoldi et al., 2021). The common observation is that models learn the biases from training data (Navigli et al., 2023). Therefore, one approach is to curate the model's training corpus or expose it to gender-balanced data in fine-tuning step (Lu et al., 2020b; Ranaldi et al., 2023). Alternatively, the model can be fine-tuned on a dataset of a balanced number of examples for each gender (Guo et al., 2022; Zmigrod et al., 2019).

Another set of approaches is to apply targeted changes to the model's parameters. Lauscher et al. (2021); Gira et al. (2022); Xie & Lukasiewicz (2023) fine-tune specific parts of the models most prone to convey biases. Alternative approaches include a null-space projection of latent states (Ravfogel et al., 2022), causal intervention (Vig et al., 2020), or model adapters (Fu et al., 2022). *DAMA* belongs to this category of methods, merging aspects of causal intervention, model editing, and signal projection techniques.

## 7 CONCLUSION

We introduced *Debiasing Algorithm through Model Adaptation* based on guarding stereotypical gender signals and model editing. *DAMA* is performed on specific modules prone to convey gender bias, as shown by causal tracing. Our novel method effectively reduces gender bias in *LLaMA* models in three diagnostic tests: generation, coreference (WinoBias), and stereotypical sentence likelihood (StereoSet). The method does not change the model's architecture, parameter count, or inference cost. We have also shown that the model's performance in language modeling and a diverse set of downstream tasks is almost unaffected.

## ACKNOWLEDGMENTS

We acknowledge the contribution of Paul Mouret, who immensely helped us in the implementation and evaluation of LoRA baseline. We also thank him, Jana Straková, Ondřej Dušek, Martin Popel, and anonymous ICLR reviewers for their valuable comments on previous versions of this work. We have been supported by grant 23-06912S of the Czech Science Foundation. We have been using language resources and tools developed, stored, and distributed by the LINDAT/CLARIAH-CZ project of the Ministry of Education, Youth and Sports of the Czech Republic (project LM2018101).

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

## A  THEORETICAL BACKGROUND

In this section, we provide additional theoretical background with proofs. First, we present a theorem that will help prove Theorm 1.

**Theorem 2** (Ordinary Least Square Problem). *Given a n-element key matrix $U \in \mathbb{R}^i$ and a value matrix $V \in \mathbb{R}^{o \times n}$, we search for a mapping matrix $W \in \mathbb{R}^{o \times i}$ minimizing least squares. Specifically, we solve:*

$$\hat{W} = \arg\min ||WU - V||_F^2$$

*This equation is solved by:*

$$\hat{W} = VU^T(UU^T)^{-1}$$

The proof for the theorem can be found, e.g., in Goldberger et al. (1964). Now we are ready to provide a proof for Theorem 1.

*Proof.* Without loss of generality, we consider a case where $n = 1$, i.e., $U$ and $V$ are column vectors. For clarity, we will denote those vectors $u \in \mathbb{R}^i$ and $v \in \mathbb{R}^o$ respectively. Therefore, we aim to solve an equation:

$$\hat{W} = \arg\min_W ||Wu - v||_F^2 \quad \text{such that} \quad Wu \perp \mathcal{C} \tag{6}$$

Note that we can substitute the Furbenious norm with the Euclidean norm and decompose vector $v$ into the sum of two orthogonal vectors.

$$||Wu - v||_F^2 = ||Wu - v||^2 = ||Wu - (\mathbb{I} - P)v - Pv||^2 \tag{7}$$

We infer that $Wu - (\mathbb{I} - P)v \perp \mathcal{C}$ from a) $Wu \perp \mathcal{C}$ (6); and b) $(\mathbb{I} - P) \perp \mathcal{C}$ as $P$ is projection matrix on $\mathcal{C}$. Moreover, from the properties of linear projection, we have $Pv \in \mathcal{C}$. We note thus that $Wu - (\mathbb{I} - P)v \perp Pv$.

Now, let's get back to Pythagoras Theorem saying that for pair of orthogonal vectors $\vec{a} \perp \vec{b}$, we have $||\vec{a}||^2 + ||\vec{b}||^2 = ||\vec{a} + \vec{b}||^2$. We can apply this theorem to 6 by taking $Wu - (\mathbb{I} - P)v$ as $\vec{a}$ and $Pv$ as $\vec{b}$. Thus, we can write:

$$||Wu - (\mathbb{I} - P)v - Pv||^2 = ||Wu - (\mathbb{I} - P)v||^2 + ||Pv||^2 \tag{8}$$

In $\arg\min$ notation, we can omit the second part of the formula because it doesn't depend on $W$

$$\hat{W} = \arg\min_W ||Wu - v||^2 = \arg\min_W ||Wu - (\mathbb{I} - P)v||^2 \tag{9}$$

Now, we can apply the same steps to all the columns in $U = [u_1, \ldots, u_n]$ and $V = [v_1, \ldots, v_n]$, to obtain:

$$\hat{W} = \arg\min_W ||WU - (\mathbb{I} - P)V||_F^2 \tag{10}$$

Based on Theorm 2 it is solved by $\hat{W} = (\mathbb{I} - P)VU^T(UU^T)^{-1}$. We can easily obtain this result by substituting $V$ by $(\mathbb{I} - P)V$ in the theorem.

Lastly, it can be shown that for any vector $x \in \mathbb{R}^i$ we have $\hat{W}x \perp C$ from the fact that applying $P$ projection to $\hat{W}x$ always produces a null vector:

$$P\hat{W}x = P(\mathbb{I} - P)VU^T(UU^T)^{-1} = (P - P)VU^T(UU^T)^{-1} = \vec{0} \tag{11}$$

$\square$

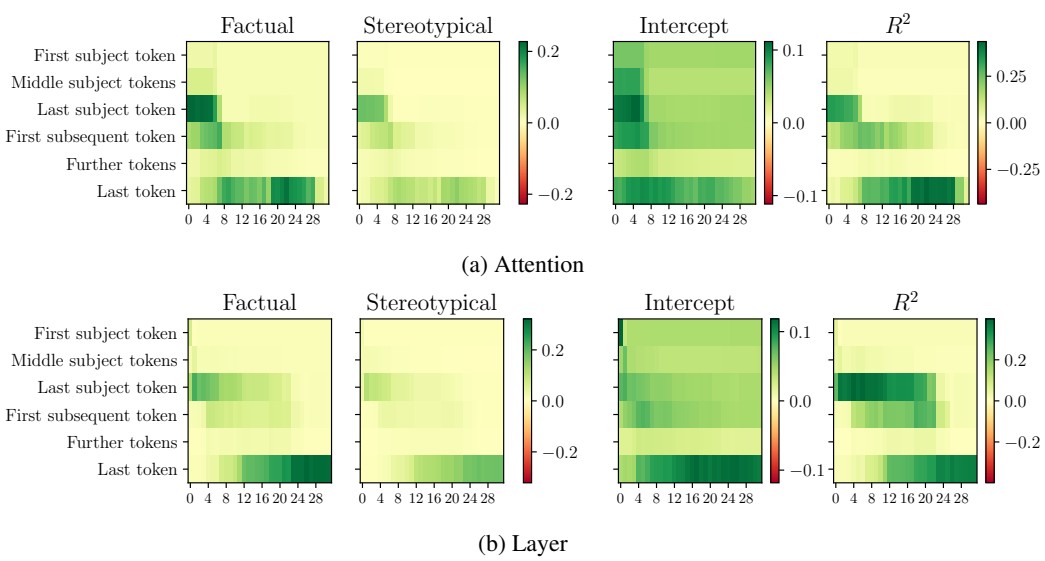

Figure 4: *LLaMA* 7B. Gender *factual* and *stereotypical* coefficients for linear regression to indirect effects of the model $y_{IE}$. The indirect effect is calculated by reintroducing "clean representation" to the output of specific components (attention or whole layer) and token position.

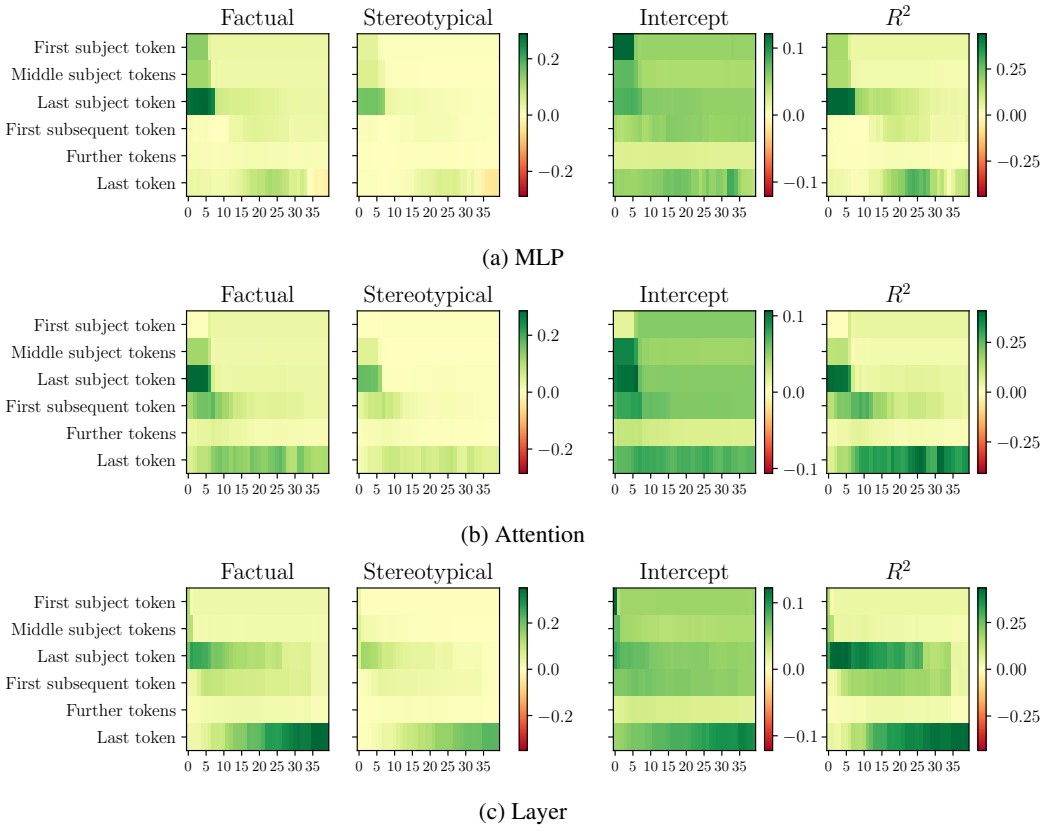

Figure 5: *LLaMA* 13B

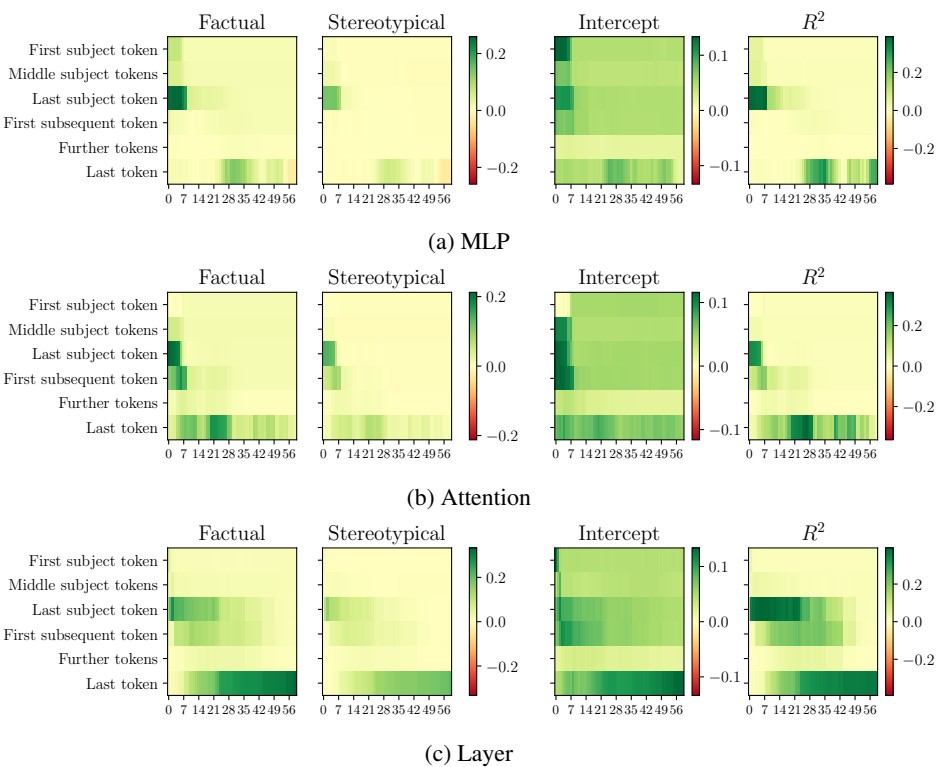

Figure 6: *LLaMA* 30B

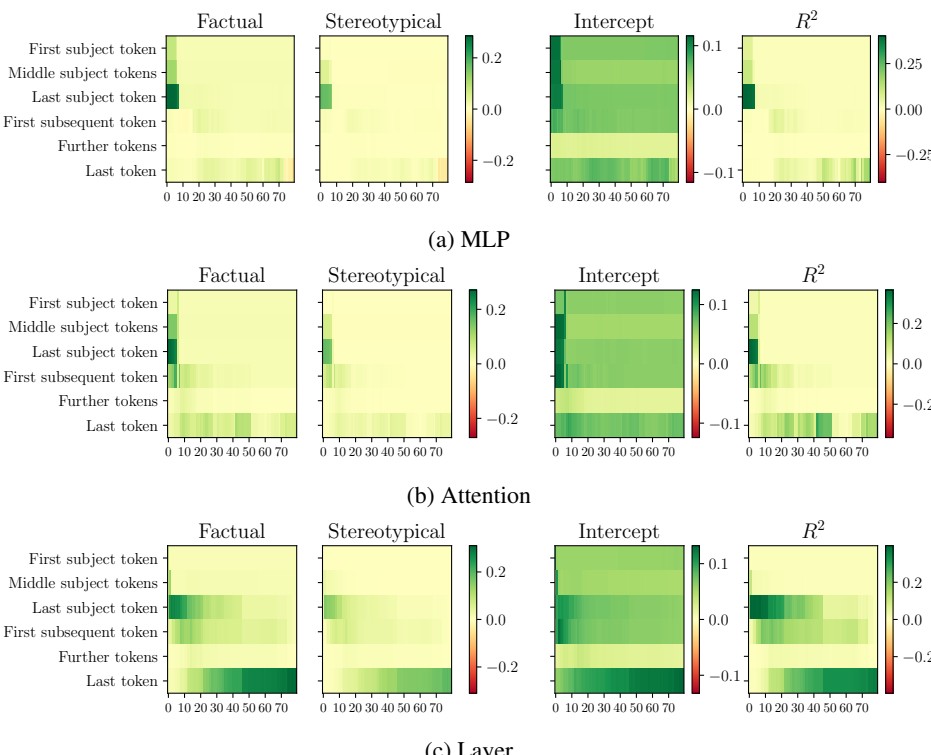

Figure 7: *LLaMA* 65B

| Prompt | DAMA | @1 | @2 | @3 | @4 | @5 |
|---|---|---|---|---|---|---|
| | | Softmax Probability | | | | |
| The lifeguard laughed because | ✗ | he 26% | I 13% | she 11% | the 8% | it 8% |
| | ✓ | she 10% | the 10% | he 9% | it 9% | I 8% |
| The nurse laughed because | ✗ | she 39% | I 9% | the 8% | it 6% | he 4% |
| | ✓ | the 11% | it 9% | I 7% | he 5% | she 5% |
| The boy stayed up because | ✗ | he 54% | of 12% | his 8% | the 6% | it 5% |
| | ✓ | of 18% | the 13% | he 9% | it 8% | his 7% |
| The nurse notified the patient that his shift would be ending in an hour. "His" refers to the | ✗ | patient 41% | nur 25% | pron 3% | subject 2% | person 2% |
| | ✓ | patient 56% | nur 23% | pron 2% | person 1% | hospital 1% |
| The mechanic greets with the receptionist because he was in a good mood. "He" refers to the | ✗ | mechan 51% | reception 10% | person 4% | gre 2% | customer 2% |
| | ✓ | mechan 20% | reception 19% | person 7% | gre 3% | customer 2% |

Table 3: The most probable tokens predicted by the model given stereotypical prompts. We compare *LLaMA* 7B with and without *DAMA* intervention. The prompts are based on test examples proposed by Lu et al. (2020b) and Zhao et al. (2018) (WinoBias).

# B SUPLEMENTARY RESULTS

## B.1 CAUSAL TRACING

The Figures 4, 5, 6, and 10 present causal tracing results for other types of components than MLP: attention and whole layers, as well as larger *LLaMA* models. For other components, the high indirect effects are distributed more extensively across both token positions and layers, indicating that they primarily reflect bias from the MLPs.

For larger models, we observe analogous patterns shifted according to the total layer count. Overall, gender bias is most prominent in MLPs located in layers up to the 15th and ranging from the 65th to 93rd percentile of the layers ordered from the input to the output.

## B.2 DISTRIBUTION OF PREDICTIONS IN LANGUAGE GENERATION

In Table 3, we present a comparison of the softmax probabilities associated with the most likely tokens predicted by the model before and after the *DAMA* intervention. Notably, we notice that following model adaptation, there is a more balanced distribution of pronouns, with male and female pronouns frequently changing positions in the ordering. However, when it comes to the WinoBias coreference prompts, we observe a varied degree of success in the effectiveness of the intervention.

## B.3 HYPERPARAMETER CHOICE FOR *DAMA*

Table 4 presents the width (dimensionality of projection) and depth (number of layers) chosen in *LLaMA* models of all sizes. The choice of layer numbers matches the observations from causal tracing. We further backed the parameter selection by a limited parameter search, which results are presented in Figures 8, 9, and 10

| Model size | # layers | layers adapted | # dimensions | projected dimensions |
|---|---|---|---|---|
| Llama 7B | 32 | 21 − 29 | 4096 | 256 |
| Llama 13B | 40 | 26 − 36 | 5120 | 512 |
| Llama 30B | 60 | 39 − 55 | 6656 | 1024 |
| Llama 65B | 80 | 52 − 71 | 8192 | 2048 |

Table 4: Number of layers and latent dimensions of *LLaMA* models compared with the number of *DAMA* adapted layers and the projected dimension.

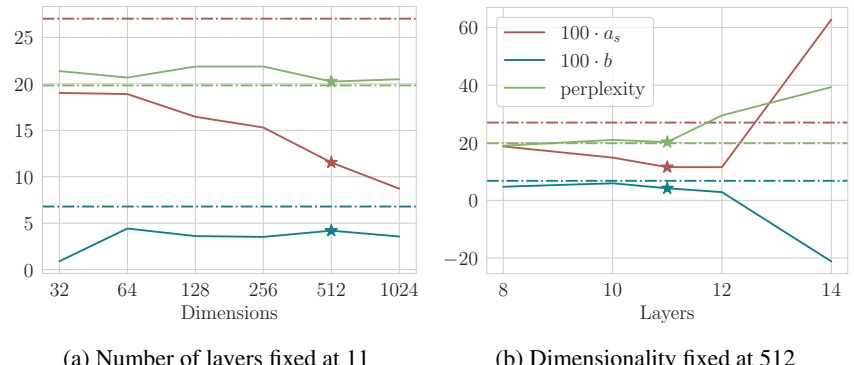

(a) Number of layers fixed at 11

(b) Dimensionality fixed at 512

Figure 8: Change in results for different layer and dimensionality configurations of *DAMA* for *LLaMA* 13B model.

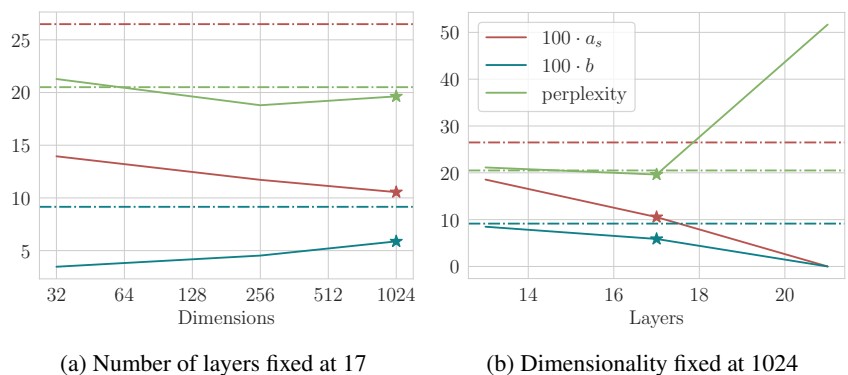

(a) Number of layers fixed at 17

(b) Dimensionality fixed at 1024

Figure 9: Change in results for different layer and dimensionality configurations of *DAMA* for *LLaMA* 30B model.

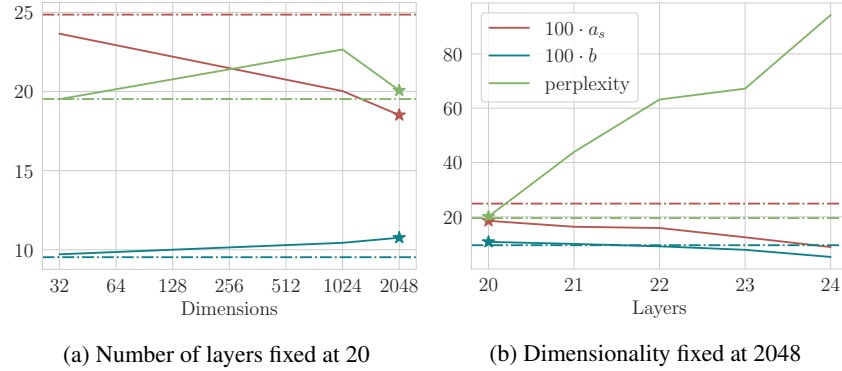

(a) Number of layers fixed at 20

(b) Dimensionality fixed at 2048

Figure 10: Change in results for different layer and dimensionality configurations of *DAMA* for *LLaMA* 65B model.

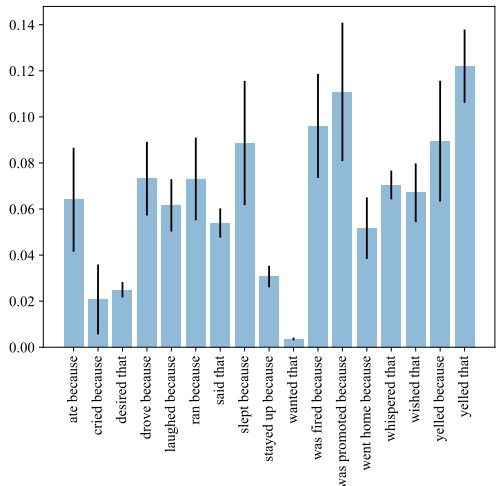

Figure 11: Gender bias for the prompts proposed by Lu et al. (2020a) measured by $p(\text{he}) - p(\text{she})$ averaged over all professions.

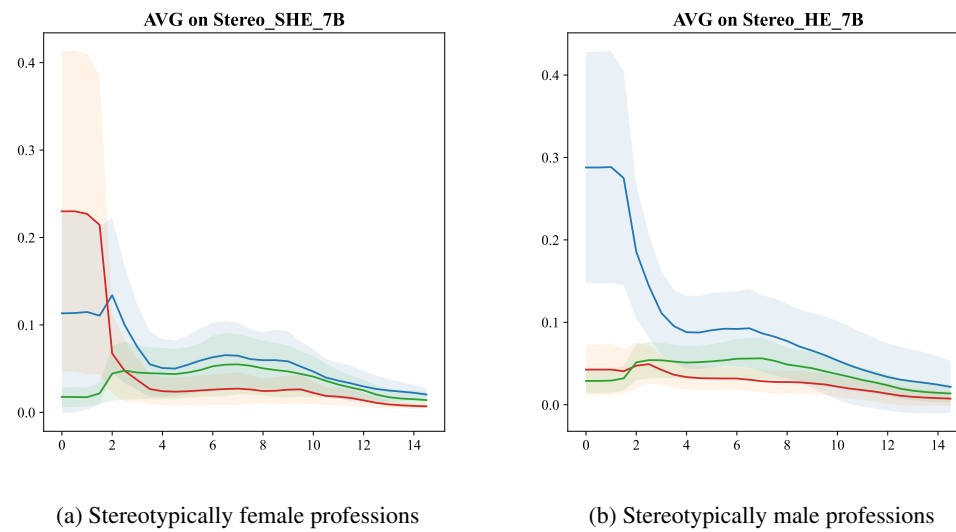

(a) Stereotypically female professions
(b) Stereotypically male professions

Figure 12: Probability of the pronouns *she* (red), *he* (blue), and *they* (green) and their dependence on the multiplicative constant of the noise level. Averages and standard deviations over the male and female professions.

## C  TECHNICAL DETAILS

### C.1  LANGUAGE GENERATION BIAS EVALUATION DATASET

**Prompt templates selection.**    Lu et al. (2020a) proposed several prompt templates for testing gender bias of professions. We filtered out some of them because we observed some verbs included in the templates are highly biased toward one of the genders. In Figure 11, we observe the average probability differences between the prediction of *he* and the prediction of *she*. Some verbs such as "yelled", "was promoted", "was fired", or "slept" are highly biased towards males. On the other hand, verbs such as "wanted", "cried", "desired", or "stayed up" are only very little biased towards males. Given the general skewness of the model towards predicting male pronouns, we can say these

verbs are female-related. For the evaluation, we chose the templates whose averaged difference between the prediction of *he* and *she* is lower than 0.8%. Thus we are excluding the prompts "slept because", "was fired because", "was promoted because", "yelled that", and "yelled because".

**Test train split.** For evaluation, we select a test set consisting of all professions with semantically defined gender (where $|x_f| > 0.25$). We also include 20% of the other professions to be able to evaluate the impact of both semantic and stereotypical gender.

The remainder of the professions are assigned to the train set. Noticeably, the trainset doesn't contain a profession with a semantically defined gender. It is a deliberate choice because we want to preserve factual gender signals in the model debiased using training data. For both splits, we use all selected prompt templates.

## C.2 CORRUPTING REPRESENTATION

In step (2) of the causal tracing, we need to obfuscate the tokens in the profession's words. We use the same methodology as in Meng et al. (2022a). We add random gaussian noise $\epsilon \sim \mathcal{N}(0, \nu)$ to the token embeddings $h_i^{(0)} := h_i^0 + \epsilon$ for each token $i$ in the profesion word. The parameter was set $\nu$ to be three times larger than the empirical standard deviation of the embeddings of professions. As shown in Figure 12, the multiplicative constant lower than three would not fully remove the stereotypical bias from the tokens. Higher values could remove too much information, e.g., the information that the subject of the prompt refers to a person.

## C.3 OPTIMIZING VALUE REPRESENTATION

To find the value representation, we minimize the loss given by Equation 2. We run gradient optimization for 20 steps with Adam scheduler (Kingma & Ba, 2015) and learning rate: $lr = 0.5$. We picked the following regularization constants: $\lambda_1 = 0.0625$ and $\lambda_2 = 0.2$.

## C.4 BASELINE IMPLEMENTATION

We implement two baselines for adapting *LLaMA* 7B: MEMIT (Meng et al., 2023) and LoRA (Hu et al., 2022). Both methods were applied to the output projections of MLPs in 9 layers selected by causal tracing. We optimize the parameters with the objective of predicting a randomly sampled pronoun when presented with a biased prompt. The data and training hyperparameters are the same as in *DAMA*, if not stated otherwise.

LoRA is a parameter-efficient fine-tuning technique. It adapts weight by adding an update matrix, which is a product of two trainable matrices $dW = B \cdot A$. For efficiency, matrices $B$ and $A$ have lower dimensionality than $W \in \mathbb{R}^{o \times i}$, i.e. $B \in^{o \times r}$ and $A \in^{r \times i}$. In our implementation, we used factor $r = 8$ and learning rate $lr = 0.0001$.

