# OpenReview forum: "Debiasing Algorithm through Model Adaptation"
_ICLR.cc/2024/Conference — ICLR 2024 poster_

### Official Review · Reviewer_FcGW · 2023-10-31

**Soundness:** 2 fair
**Presentation:** 1 poor
**Contribution:** 2 fair
**Rating:** 3
**Confidence:** 4

**Summary:**

This work studies the the issues of biases and stereotypes in the large language models, and propose methods to (1) identify the specific components of LLMs that is responsible for these issues; and (2) resolve these issues with a linear projection on the feed-forward components of LLMs.

**Strengths:**

- The research question is interesting and critical to LLM research and application
- The high-level ideas of the proposed techniques (on both identification and debiasing) make sense.

**Weaknesses:**

I had a difficult time understanding this paper because of the following writing issues.

- First of all, certain concepts are mentioned in the paper. However, they are either inconsistent or not explicitly defined. For example,
    - Stereotypical keys vs. stereotyped keys
    - Gender value vs. gendered value
    - Also, what is grammatical gender?
- About Equation 2, I have several questions
    - What is z? Why do we need a $\ell_2$ item on it?
    - Missing a “)” somewhere in equation 2
    - What is $P(o’|X’)$? The first two items in equations look similar to variational inference. In that case, $P(o’|X’)$ should be something similar to a prior distribution. However, I could not find its definition.
- What is gender values metrics?  What is the relation between these V metrics and U?
- Why $P$ in equation 5 is defined in that way, and how should we use it? I think it was explained in the paragraph right after equation 5, but I am not sure I understand it.
- Figure 3 (b) seems to indicate that to reduce the bias, the cost is to get a much worse language model (based on the perplexity score), which is not exactly claimed in the paper.

In addition to my clarification question, one concern is about the technical novelty specifically the causal tracing idea is from prior work, and the linear project seems to be a straightforward idea in debiasing literature, so I am wondering what is the technical novelty of this work.

**Questions:**

Please refer to the clarification questions mentioned in the previous section.

---

> ### Author Response · Authors · 2023-11-20
>
> Thank you for your detailed comments and for recognizing the importance of debiasing in LLM research. We organize the rebuttal in three parts to comprehensively answer the raised concerns related to: 1) LM performance, 2) clarifications about concepts and equations in section 4;  3) the novelty of our approach.
>
> ## Firstly, we want to address the concern about LM performance that might be an effect of misunderstanding our ablation setting:
> > Figure 3 (b) seems to indicate that to reduce the bias, the cost is to get a much worse language model.
>
>
> Figure 3 (b) illustrates ablation experiments with varying intervention layer numbers. The significant drop in perplexity occurs notably for 10 or 11 layers. It's crucial to note that, as stated in the beginning of section 4.4: "We apply DAMA to MLPs in approximately one-third of the model’s upper layers (in LLaMA 7B layers 21 - 29 out of 32),” equating to 9 layers. The corresponding perplexity is denoted with a star on the figure, which is close to the score of the original model.
>
>
> ## Secondly, we address your questions regarding concepts in Section 4. Moreover, we refined the section for better clarity and uploaded it to open review.
> > "Certain concepts [..] are either inconsistent or not explicitly defined."
>
> We clarified in section 4 that:
> - “stereotypical keys” and “stereotyped keys” are the same entity. We use just the former name.
> - “gender values” and. “gendered values” are the same entity. The latter name is now used exclusively.
> - “Grammatical gender”  is the quality manifested by the specific inflection of words. We now use a descriptive definition of grammatical gender to avoid confusion with gender scores.
>
> > Questions regarding Equation 2
>
> $z$ is the latent vector, specifically the output of the lth feed-forward layer, as denoted in the subscript  $FF_{out,l}=z$. We used it just as a loss argument to find the gendered value vector $v_o$
>  We added $\text{argmin}$ notation in Equation 2 to clarify that. The missing parenthesis was also fixed.
>
> > "The first two items in equations look similar to variational inference."
>
> In equation 2, we consider the model's output probability $P(o'|X')$  conditioned on a known prompt ( $X'$). The formula resembles variational inference, yet there is a crucial difference: $X'$ is a known prompt, not a random variable. As a result, a prior distribution cannot be defined here except for a trivial one.
> > Meaning of metrics $U$,$V$, and $P$:
>
> We just want to clarify that $U$ and $V$ are matrices of concatenated stereotypical key vectors and gendered value vectors as described in the article (those are not metrics). We added further explanation regarding equation 5 in the revised paper.
>
> ## Lastly, we’d like to highlight the novelty of DAMA presented in the Discussion Section.
>
> > Novelty in the context of model editing methods
>
> Previous model editing methods targeted highly specific information encoded in the model ( Mitchell et al., 2022; De Cao et al., 2021), while we introduced a method capable of editing general dataset artifacts, such as various manifestations of gender bias. In Table 2, we show that DAMA significantly outperforms MEMIT (Meng et al., 2022 ) in this aspect.
>
> >  Advancements in projection-based debaising.
>
> In contrast to earlier projection-based debiasing methods that focused on learning and projecting latent representations for specific gender-related tasks (e.g., gender classification in bios Ravfogel et al. 2020), our method demonstrates that projection can be achieved without relying on auxiliary tasks. Notably, the projections in our approach are directly applied to the model component, avoiding alterations to the architecture or parameter size. Our theoretical and empirical evidence further establishes that applying debiasing projections on MLPs effectively reduces bias while preserving other encoded information, showcasing the generalizability of the method across tasks.
>
>
> We hope that the response addresses most, if not all, of your concerns. We are open to further discussion, suggestions, or questions you may have.

---

### Official Review · Reviewer_2cbg · 2023-11-08

**Soundness:** 3 good
**Presentation:** 4 excellent
**Contribution:** 3 good
**Rating:** 6
**Confidence:** 3

**Summary:**

The paper demonstrated unwanted gender bias still exists in the popular LLaMA family of models. To combat these prevalent biases, the paper located components of the model that caused such biases and edited these weights to mitigate the effect of gender biases on downstream tasks. The method, DAMA, is an improvement over the previous method both in terms of reduction rate and maintaining original performances.

**Strengths:**

1. Detailed and quantitative measurement of the effect of factual cues and stereotypical cues on model generations. Specifically in the result section, the authors measured the effect of factual cues and stereotypical cues based on layer number and token positions.
2. The gender bias reduction method via model weight editing has sound theoretical backup.
3. DAMA effectively reduces the bias rate without hurting too much of the downstream performances.

**Weaknesses:**

1. Seems like there is still quite some room for improvements in the debiasing methods on all three evaluations proposed in the paper. For WinoBias and SteroSet gender, I interpret the results as being still pretty gender biased even after applying DAMA. And for Bias in LM, for larger models, the a_s and a_f are still far from complete removal. It would help greatly to provide other gender bias removal methods as baselines to better assess how well DAMA did.
2. I think the study on the effects of downstream is great. However, its negative effect on reasoning tasks such as ARC can be concerning for a practical user. Is there any hypothesis on why it doesn't fare well with reasoning type of tasks?

**Questions:**

Styling:
1. You are missing a right parenthesis on equation (2) for the KL divergence.

---

> ### Author Response · Authors · 2023-11-20
>
> Thank you for your insightful review, particularly for acknowledging the significance of evaluating diverse gender bias metrics and downstream tasks. We also appreciate your positive evaluation of our method's theoretical background.
>
>
> > Seems like there is still quite some room for improvements
>
> We agree that the method can be further improved for larger scales. In our opinion, the crucial strength of DAMA is its versatility in reducing bias for professions and tasks unseen in the adaptation. This contrasts with many past works that did not show consistent patterns across tasks (Delobelle et al. 2022, van der Wal et al. 2023). Admittedly,  bias annotation is oftentimes not consistent. Thus, some manifestation of it may always be present in the model when not specifically targeted.
>
>
> > Is there any hypothesis on why it doesn't fare well with the reasoning type of tasks? ARC
>
> Please note that the difference between model instances in Table 2 is only barely statistically significant.
> We compared the errors on the ARC-C task made by the original 65B model and the same model with DAMA and did not observe any clear patterns. On average, the original model selects the correct answer for 40% of DAMA’s errors, while DAMA picks the correct answer for 33% of the original model’s errors. Considering the probability of randomly choosing the correct answer is 33%, it suggests that the performance difference may be attributed to a random factor.
>
> > You are missing a right parenthesis on equation (2) for the KL divergence.
>
> The equation was fixed in the updated version of the paper.
>
> Thank you once more for your review. Should you have any additional comments or questions,  we'll be happy to address them.

---

### Official Review · Reviewer_S9AC · 2023-11-10

**Soundness:** 3 good
**Presentation:** 3 good
**Contribution:** 3 good
**Rating:** 8
**Confidence:** 3

**Summary:**

The paper investigates gender bias in the Llama model. First, an existing causal tracing method is adapted to work for measuring gender bias in different components of the model. Then, a method for modifying the model weights is described for reducing the gender bias. Evaluation is performed both on gender bias benchmarks and downstream task datasets.

**Strengths:**

The methods are interesting. Evaluation is performed on a number of different datasets and using different measures.

The method for updating the model is also interesting. Particularly as it manages to not even increase the number of parameters in the model.

**Weaknesses:**

A weakness is only focusing on Llama, as it is unclear how much these findings generalise to other models.

The causal tracing method is interesting. It is currently unclear how much this is based on previous work and what exactly is the novel contribution. Please clarify this.

It is unclear why a linear model is fitted across the two gender scores in order to investigate the extent of bias. If the two scores are correlated (which they likely are) then the coefficients of such a linear model might not give accurate indications of the different biases. Measuring correlation with the gender scores seems like a much more straightforward method.

Again, for the method of updating weights, please clarify the difference of the proposed method to previous work, including Rafvogel et al 2022.

A major selling point of the method seems to be that the proposed DAMA still achieves good performance on downstream tasks. However, the baseline MEMIT actually seems to get better results on most of this metrics. This finding somewhat weakens this claim and should be addressed in the paper.

**Questions:**

Please see above

---

> ### Author Response · Authors · 2023-11-20
>
> Thank you for your thoughtful comments and positive review of our work and for recognizing the importance of the proposed approach. We appreciate your acknowledgment of the need to evaluate the debiased model on a wide range of bias metrics and downstream tasks.
>
> In this response, we want to highlight the novelty of DAMA and further explain our choices in the experimental setting.
>
> ## Novelty of DAMA
>
> > Novelty in the context of model editing methods and comparison to MEMIT
>
> Previous model editing methods targeted highly specific information encoded in the model ( Mitchell et al., 2022; De Cao et al., 2021), while we introduced a method capable of editing general dataset artifacts, such as various manifestations of gender bias.  Specifically, we show that the intervention performed on a set of professions generalizes well to unseen professions and even other manifestations of Gender Bias. This is beyond the capabilities of MEMIT (Meng et al. 2022), which targets only specific pieces of information that are sparsely encoded. Hence, MEMIT intervention shows marginally better performance on downstream tasks while still keeping bias in the model.
>
>
> >  Advancements in projection-based debaising.
>
> In contrast to earlier projection-based debiasing methods that focused on learning and projecting latent representations for specific gender-related tasks (e.g., gender classification in bios Ravfogel et al. 2020), our method demonstrates that projection can be achieved without relying on auxiliary tasks. Notably, the projections in our approach are directly applied to the model component, avoiding alterations to the architecture or parameter size. Our theoretical and empirical evidence further establishes that applying debiasing projections on MLPs effectively reduces bias while preserving other encoded information, showcasing the generalizability of the method across tasks.
>
> ## Further explanation for experimental setting choices:
>
> > It is unclear why a linear model is fitted across the two gender scores in order to investigate the extent of bias.
>
> Thank you for this insightful observation. We have also checked the linear coefficients of the linear models fitted on one of the gender scores (factual and stereotypical). Notably, the patterns are closely aligned with those observed in the joint linear model, visible in the table below.  We opted to present the coefficients of a joint model in the paper because it has one intercept coefficient, which has a clear interpretation as the language model’s preference for male pronoun when the subject is marginalized (as noted at the end of section 2.3).
>
> |          | Joint  |       | Separate |       |
> |----------|--------|-------|----------|-------|
> |          | $a_s$  | $a_f$ | $a_s$    | $a_f$ |
> | Original |  0.235 | 0.320 |    0.325 | 0.371 |
> | w MEMIT  |  0.209 | 0.282 |    0.288 | 0.327 |
> | w DAMA   | -0.005 | 0.037 |   -0.006 | 0.035 |
>
> Moreover, Pearson’s correlation between factual and stereotypical scores is moderately low: $\rho=0.258$ yet statistically significant.
>
> > only focusing on Llama, as it is unclear how much these findings generalize to other models.
>
> We intend to explore the applicability of DAMA with other models and languages in future work. We chose LLaMA because of its strong performance and availability of weights across different scales.
>
> Once again, thank you for your review. We will be happy to answer any further questions regarding our method.

---

### Meta-Review · Area_Chair_EebK · 2023-12-07

**Metareview:**

This paper presents a modeling algorithm to improve the gender bias of LLMs. The authors specifically looks at the projection layer of the model,  perform analysis showing that the model parameters are inherently biased in favor of one gender over another. This is expected as this is the kind of data the LLMs are trained on. The analysis sheds light on this problem and then the authors present a method that overcomes this issue by mutating the weights of the projection layer without introducing any new model parameters. This is interesting and different from the previous approaches. The authors then show improvement on standard bias datasets while also showing that the performance on the debiased model is not significantly reduced on the general language understanding tasks. Overall this is a nice contribution that can be applied post-training to models.

**Justification For Why Not Higher Score:**

There is still significant headroom in how much more improvements can be obtained over the results achieved by this method.

**Justification For Why Not Lower Score:**

I think it is a good technique for debiasing which is significantly different from approaches presented in the past and comes with rigorous experimentation.

---

### Decision · Program_Chairs · 2024-01-16

Accept (poster)